# Real-Time EEG-Based Emotion Recognition

**DOI:** 10.3390/s23187853

**Published:** 2023-09-13

**Authors:** Xiangkun Yu, Zhengjie Li, Zhibang Zang, Yinhua Liu

**Affiliations:** 1College of Computer Science and Technology, Qingdao University, Qingdao 266071, China; 2School of Automation, Qingdao University, Qingdao 266071, China; 3Shandong Key Laboratory of Industrial Control Technology, Qingdao 266071, China; 4Institute for Future, Qingdao University, Qingdao 266071, China

**Keywords:** EEG, emotion recognition, affective computing, real-time

## Abstract

Most studies have demonstrated that EEG can be applied to emotion recognition. In the process of EEG-based emotion recognition, real-time is an important feature. In this paper, the real-time problem of emotion recognition based on EEG is explained and analyzed. Secondly, the short time window length and attention mechanisms are designed on EEG signals to follow emotion change over time. Then, long short-term memory with the additive attention mechanism is used for emotion recognition, due to timely emotion updates, and the model is applied to the SEED and SEED-IV datasets to verify the feasibility of real-time emotion recognition. The results show that the model performs relatively well in terms of real-time performance, with accuracy rates of 85.40% and 74.26% on SEED and SEED-IV, but the accuracy rate has not reached the ideal state due to data labeling and other losses in the pursuit of real-time performance.

## 1. Introduction

Affective computing represents a relatively novel research domain that addresses the challenge of enabling computer systems to accurately process, recognize, and understand human-expressed emotional information for natural human–computer interaction, making it of paramount importance in the field of artificial intelligence [1,2]. The advancement of affective computing requires interdisciplinary efforts, and its research outcomes contribute to progress in various fields, including computer science and psychology. Emotion recognition is a facet of affective computing that is increasingly attracting the attention of researchers with interdisciplinary backgrounds. Emotions, as psychological and physiological states accompanying cognitive and conscious processes, play a crucial role in human society. The emergence of artificial intelligence has laid the foundation for affective-aware artificial intelligence, aiming to imbue machines with emotions and facilitating enhanced human–computer interaction beyond previous isolated interactions. Therefore, the realization of artificial intelligence hinges on achieving affective-aware artificial intelligence, with a pivotal focus on endowing artificial intelligence with the capability to perceive, understand, and regulate emotions.

There are various methods for recognizing emotions, including both non-physiological and physiological signals. Non-physiological signals include facial expressions and body posture, which are used in psychology to infer a person’s internal emotional state. Non-physiological signals offer the advantage of being easily accessible, but they also have limitations. For example, facial expressions can be deceptive; as seen in cases of fraud, individuals can maintain their original expressions even when faced with evidence, making it challenging to discern their inner emotional activity solely from their facial expressions. Acquiring physiological signals is more complex but yields greater accuracy. There are several types of physiological signals, such as electroencephalography (EEG) and electrocardiography (ECG). Among these, EEG is widely used for emotion detection [3,4]. EEG captures signals directly from the surface of the cerebral cortex, offering a direct and comprehensive means to recognize emotions with excellent temporal resolution [5,6,7]. Compared to other detection methods, EEG has advantages including simplicity, portability, ease of use, and non-invasiveness. Compared to other physiological signals like ECG, skin temperature, and skin conductance, EEG signals can achieve higher classification accuracy [8]. Furthermore, EEG detects the actual response of the brain, rather than relying on subjective facial expressions, making it a more objective measure.

To recognize emotions using EEG, it is necessary to first quantify emotional states. Currently, there are two types of emotional models based on EEG: discrete models and dimensional models. Discrete models consist of a limited number of discrete basic emotions, such as happiness, sadness, surprise, and fear. The SEED-IV dataset is a commonly used discrete model EEG emotion recognition dataset, which includes four emotions: neutral, happy, sad, and fearful. Dimensional models mainly refer to the valence and arousal dimensions. Valence represents the positive or negative aspect of an emotion, while arousal represents its intensity. Dominance and liking are also used as supplementary dimensions. Dominance refers to the degree of control of the emotion, while liking indicates the degree of pleasure associated with the emotion. The DEAP dataset, widely used for EEG emotion recognition, uses these four dimensions as emotion rating standards.

There are many aspects of emotion recognition based on EEG that can be studied, and in this paper, we focus mainly on the real-time aspect of EEG emotion recognition. As mentioned earlier, EEG has good temporal resolution [5,6,7], so real-time processing is an important research area for EEG emotion recognition. Significant achievements have been made in previous studies on real-time emotion recognition. Viet Hoang et al. [9] developed a real-time emotion recognition system to identify two valence classes and two arousal classes, resulting in four basic emotions (happiness, relaxation, sadness, and anger) and neutral state combinations. The average accuracy of emotion recognition for all subjects was 70.5% (ranging from 66% to 76%). W.-C. Fang et al. [10] proposed a real-time EEG-based emotion recognition hardware system architecture based on a multiphase convolutional neural network (CNN) algorithm. In this work, six EEG channels (FP1, FP2, F3, F4, F7, and F8) were selected, and EEG images were generated from the fusion of spectrograms. The average accuracy of valence and arousal for the subjects was 83.36% and 76.67%, respectively. J. W. Li et al. [11] proposed a technique called Brain Rhythm Sequencing (BRS), which interprets EEG signals based on dominant brain rhythms with maximum instantaneous power at each 0.2-s timestamp. Results from Music Emotion Recognition (MER) experiments and three emotion datasets (SEED, DEAP, and MAHNOB) demonstrate that the classification accuracy of single-channel data with a duration of 10 s ranges from 70–82%. Li, Z. et al. [12] proposed an improved feature selection algorithm based on EEG signals for recognizing participants’ emotional states, and designed an online emotion recognition brain–computer interface (BCI) system combining this feature selection method. Results show that the average accuracy for four-level emotion recognition reached 76.67%. Y. -J. Liu et al. [13] proposed a real-time movie-induced emotion recognition system that identifies individuals’ emotional states by analyzing their brain waves. Overall accuracy reached 92.26% for identifying high arousal and positive emotions from neutral, and 86.63% for identifying positive emotions from negative emotions.

Based on the above research, the work of this paper is as follows:(1)The real-time problem of emotion recognition based on EEG is explained, and the problem is fully dissected.(2)Aiming at the real-time problem, it is proposed to find the application of the appropriate time window length to improve the real-time performance, and many experiments are carried out to find it.(3)In order to prove the real-time performance, the model combined with LSTM and the attention mechanism is used to conduct experiments on SEED and SEED-IV datasets, and compare with other methods.

## 2. Problem Statement

Emotion is continuous and variable [14], which makes emotion recognition difficult during emotion changing. EEG is an electrical signal that reflects brain activity. It is real-time and can be measured quickly, which makes it suitable for capturing high-frequency emotional change.

During emotion recognition, changes in emotions should be captured as early as possible. In current research on EEG-based emotion recognition, a typical approach involves using one-minute data as a sample, with each sample corresponding to a specific emotional label. This results in only one emotion being recognized in a one-minute sample. In order to improve the recognition efficiency and identify new emotions earlier, the existing method separates the one-minute data into multiple segments of the same length. J. W. Li et al. proposed a technique called Brain Rhythms Sequencing (BRS), which achieved a classification accuracy of 70–82% for single-channel data lasting only 10 s during experimental evaluation. Due to the different duration of each emotion [15,16,17], the 10 s sample may contain multiple emotions and they cannot be fully identified, causing the 10 s segmentation to still not be suitable for real-time emotion recognition.

With the existing methods, an emotion can be recognized only when it occupies a dominant proportion within a data segment. The results focus on the predominant emotion, and real-time performance cannot be guaranteed. As shown in Figure 1, a data segment contains two emotions: sadness and happiness. Initially, sadness is recognized first, but when happiness appears, it is not identified in a timely manner due to its extremely short duration. When both sadness and happiness occupy the same account of time within the segment window, the result is random. Only when happiness exists for a sufficient time can it be recognized. There exists a time Δt between the appearance of happiness and its identification, which represents the wasted time during the recognition process. Real-time emotion recognition aims to minimize Δt as much as possible.

As mentioned above, when segmenting the sample data into multiple parts of equal length with time windows, Δt can be reduced by changing the length of the segments. An appropriate value should be selected as the length of the windows. If the length is too short, although it allows for quicker updates of emotional states, there are few features in the data related to emotions, which might be insufficient for accurate emotion recognition. If the length is too long, the recognition outcome will be affected by the averaging over longer time intervals, preventing real-time performance. Only with an appropriate length can emotion be accurately captured in real-time changes without being influenced by averaging over longer time intervals, enabling faster analysis and generation of emotion recognition results. Additionally, real-time requires timely recognition of the latest emotions, so paying more attention to the latest data is also a key problem of real-time performance. When identifying the latest emerging emotions, the weight of the new data should be increased to minimize the impact of past data on the recognition results and to ensure the accuracy of emotion recognition.

## 3. Methodology

### 3.1. Emotion Recognition Model

Addressing the real-time problems mentioned above needs a suitable EEG emotion recognition model, whose key components include the size of the time window and the sliding step. Previous research has indicated that the window length for emotion recognition should not exceed 3 s. If that is the case, many emotions whose duration is significantly shorter than 3 s may go unrecognized, which does not meet real-time requirements. Additionally, when sliding the window, the step size should not exceed the window length. An excessively large step size can lead to a loss of detailed information, resulting in inaccurate results. The step size should be set at 50% to 80% of the window length to preserve more emotional variation information and short-term features in the signal and to reduce edge effects.

For emotion recognition models, using attention mechanisms can significantly improve recognition ability by weighting data. When identifying continuous signal data within sliding time windows, more attention should be given to the emotion that just appeared at the end of the time window. The use of attention mechanisms is to achieve the coordination of data weights between the past and future. In recent years, attention mechanisms have shown great performance in various fields such as image processing and natural language processing [18]. Among several attention mechanisms, we choose to use additive attention. It calculates attention weights by learning the similarity between different time points in the EEG signal sequence. During weight calculation, key vectors and query vectors at each time point are weighted and combined, and the result is used as the numerator of the attention weight. This mechanism allows learning, weighting, combining, and outputting of key information in the EEG signal sequence.

The modeling approach should incorporate EEG spatial, temporal, and frequency features. As mentioned above, focusing on the emergence of new emotions requires the model to possess the ability to combine attention mechanisms. Meanwhile, recognizing emotions in new data depends on the past data, so the model should retain and access previous data. EEG data are noisy and contain various interference signals, such as muscle movements. In addition to preprocessing the data to reduce interference, the model must be capable of handling noise and outliers. EEG detection devices are multi-channel, which allows obtaining more accurate and detailed information. Thus, the model needs to preserve the relationship between all EEG channels. In summary, we introduce LSTM to achieve this.

We propose a model suitable for EEG real-time emotion recognition, as shown in Figure 2. This model uses shorter time windows to capture detailed information and focuses on emotional changes. It includes the processing module, STFT module, modeling module, and recognition module. Firstly, in the processing module, the data are downsampled to 200 Hz, and a band-pass filter is applied to remove signals outside the 8–45 Hz range. In the STFT module, the data are processed using the Short-Time Fourier Transform (STFT) to segment and extract features from each EEG signal channel. The processed data are then input to the modeling module. The data pass through the LSTM layer first and then enter the attention layer. After obtaining the output from the attention layer, it is weighted, fused with the output from the LSTM layer, and subjected to flattening operations. Finally, the recognition results are obtained through the softmax layer of the recognition module.

### 3.2. Feature Extraction

Feature extraction is a crucial step in EEG-based emotion recognition. Extracting representative and discriminative features from EEG signals facilitates subsequent classification and recognition. During the recognition process, both temporal and spectral features need to be extracted since the features of EEG signals vary over time. Additionally, it is essential to perform feature extraction separately for each EEG signal channel without concatenating the extracted features, to maintain the independence of each channel and preserve more detailed information at every time point. To achieve these objectives, we employ the Short-Time Fourier Transform (STFT) for feature extraction. By applying time-domain windowing to EEG signals and performing frequency-domain Fourier Transform on the windowed signals, we obtain the spectral distribution of EEG signals at various time instances and frequencies, revealing the short-term frequency change characteristics of the EEG signals. STFT is a commonly used method for time-frequency analysis, which decomposes signals into short-time frequency components. The formula is as follows:(1)$$F\left(\tau ,\omega \right)=\int_{-\infty }^{\infty }f\left(t\right)w\left(t-\tau \right)e^{-jwt}dt$$
where f(t) is the input signal and w(t−τ) is a window function. In previous research, different window lengths have been used for feature extraction in EEG signal processing. Ouyang et al. [19] studied various window lengths for EEG-based emotion recognition and found that the optimal window length for emotion recognition is 1–2 s. We conduct multiple experiments using different time window lengths and ultimately select a one-second time window with a 50% overlap.

### 3.3. LSTM with Attention Mechanism

After careful consideration, we adopt a network model that combines LSTM with the additive attention mechanism. This model demonstrates excellent capabilities in handling time-series data, such as EEG signals, allowing itself to sequentially capture temporal information from the input signal data and adapt well to the characteristics of EEG signals.

LSTM [20] is a variant of RNN [21] and is known for effectively processing sequential data. Its effectiveness in extracting temporal information from biological signals has been demonstrated. LSTM comprises cell states that propagate and store temporal information over time, as well as input and output gates. The formulas of LSTM are as follows:(2)$$i_{t}=\sigma \left(W_{xi}x_{t}+W_{hi}h_{t-1}+b_{i}\right)$$
(3)$$o_{t}=\sigma \left(W_{xo}x_{t}+W_{ho}h_{t-1}+b_{o}\right)$$
(4)$$f_{t}=\sigma \left(W_{xf}x_{t}+W_{hf}h_{t-1}+b_{f}\right)$$
(5)$$\tilde{C_{t}}=\tanh\left(W_{xc}x_{t}+W_{hc}h_{t-1}+b_{c}\right)$$
(6)$$C_{t}=f_{t}⊙C_{t-1}+i_{t}⊙\tilde{C_{t}}$$
(7)$$h_{t}=o_{t}⊙\tanh\left(C_{t}\right)$$

Among them, $W_{xi}$, $W_{hi}$, $b_{i}$, $W_{xo}$, $W_{ho}$, $b_{o}$, $W_{xf}$, $W_{hf}$, $b_{f}$, $W_{xc}$, $W_{hc}$, $b_{c}$ are the learned parameters, $h_{t-1}$ is the hidden state of the previous time step, $x_{t}$ is the input of the current time step, $i_{t}$ is the input gate vector, and $o_{t}$ is the output gate Vector, $f_{t}$ is the forget gate vector, $\tilde{C_{t}}$ is the candidate cell state of the current time step, $C_{t}$ is the cell state of the current time step, $h_{t}$ is the hidden state of the current time step, σ represents the sigmoid function, tanh represents the hyperbolic tangent function, and ⊙ represents Dot multiplication.

The additive attention mechanism calculates attention weights by learning the similarity between different time points in the EEG signal sequence. During weight calculation, key vectors and query vectors at every time point are weighted and combined, and the result is used as the numerator of the attention weight. This method allows learning, weighting, combining, and outputting of key information in the EEG signal sequence.
(8)$$A=\mathrm{softmax}\left(w^{T}X\right)\in R^{n}$$
(9)$$Z=XA^{T}\in R^{d}$$

The attention weight is within the range of [0, 1], and the sum of weights is equal to 1.

## 4. Experimental Results and Discussion

### 4.1. Datasets

To validate the performance of the proposed algorithm and demonstrate the feasibility of real-time emotion recognition, we conducted experiments on the SEED [22,23] dataset and SEED-IV [24] dataset. The SEED dataset consists of EEG and eye-tracking data from 12 subjects, along with EEG data from the other 3 subjects. The data were collected while they were watching selected movie clips that elicited positive, negative, and neutral emotions. The SEED-IV dataset is an evolution of the SEED dataset, increasing the emotional categories from three to four, including happiness, sadness, fear, and neutral. Unlike the SEED dataset, SEED-IV is a multi-modal dataset designed for emotion recognition. It provides not only EEG signals but also eye movement features obtained from SMI eye-tracking glasses. These two datasets were chosen because they are based on discrete models, which are preferable for real-time emotion recognition research compared to dimensional models. Additionally, both of these datasets encompass a variety of emotions, which enhances the persuasiveness of the experimental results.

During the process of data collection, EEG signals are susceptible to external interference, resulting in significant noise within the EEG data. To mitigate the impact of the noise, it is necessary to perform denoising on the signals. The noise can originate from various sources, such as sensors, electromagnetic interference, and other unexpected signal sources. In this study, we downsampled the EEG signal data to 200 Hz. Extensive research has shown that the electrical activity of cortical neurons is relatively weak, and selecting a frequency range suitable for EEG emotion recognition is beneficial for the experiment. There are typically five types of brain waves: Delta, Theta, Alpha, Beta, and Gamma. Delta waves usually occur between 0.5 Hz to 4 Hz and are associated with deep sleep and physical recovery. They are often observed in the EEG of infants with incomplete brain development and in the deep sleep of adults with certain brain disorders. Theta waves typically occur between 4 to 8 Hz and are associated with sleep, relaxation, meditation, and creative thinking. Alpha waves usually occur between 8 Hz to 13 Hz and are mainly observed during relaxation, stillness, closing the eyes, and deep relaxation. Beta waves typically occur between 13 Hz and 30 Hz and are primarily observed during high cognitive load states, such as thinking, attention, and anxiety. Gamma waves usually occur above 30 Hz and are associated with higher cognitive functions such as learning, attention, consciousness, and perception. Considering the factors mentioned above, we used a band-pass filter to select data within the 8–45 Hz range for our experiments.

### 4.2. Experimental Setting

The experimental effects of different parameters were compared through multiple sets of experiments, and the following parameter combinations were tried:

Number of nodes: 32, 62, 128

Learning rate: 0.01, 0.001, 0.005

Dropout rate: 0.2, 0.3, 0.5

Window length: 0.5 s, 1 s, 1.5 s, 2 s

Window overlap rate: 20%, 30%, 50%

The experimental results indicate that the window length of one-second performs better. Compared to the two-second window, it can more accurately capture subtle temporal variations and dynamic changes in emotions. The shorter window length, however, leads to a reduced amount of data, resulting in information loss and an inability to better reflect the signal’s dynamic changes, making 0.5 s perform worse than 1 s. Additionally, the results also demonstrate that the 50% window overlap rate better preserves signal variations and short-term features. The network was trained using the Adam optimization algorithm with 128 nodes, the window length is 1 s, the window overlap rate is 50%, the learning rate is 0.001, the dropout rate is 0.5, and the maximum epochs is 500. Furthermore, the EarlyStopping callback function was employed, which stops the training if the validation accuracy does not increase by 0.0001 for the past 50 epochs or if the validation loss ceases to decrease.

### 4.3. Results

#### 4.3.1. Real-Time Verification

Compared to many previous studies, our experiment shows higher real-time performance. For a 60 s data segment, using a 10 s time window without overlap, we can recognize 6 instances of emotions. However, by using a 1 s time window without overlap, we can recognize 60 instances of emotions. With a 50% overlap rate, it means updating the recognition result every 0.5 s, significantly improving the efficiency of emotion recognition. As shown in Figure 3, the above figure uses the result of 5 s time window length recognition, which can only identify one major emotion; the following figure shows that when using the 1 s time window length and 50% overlap rate, while using the attention mechanism to give higher weight to the latest data, 9 emotions can be identified, greatly improving the efficiency of recognition.

#### 4.3.2. Comparison with Methods

The experimental results when using the SEED dataset are shown in Table 1. The model used in our study outperforms KNN [22] and CNN [25] in accuracy, reaching 85.40%. This clearly demonstrates the significant effect of using shorter time windows and focusing on the latest data during the emotion recognition process, showcasing the feasibility of real-time EEG emotion recognition. However, the experimental results are not as good as DGCNN [26], GLEM [27], BODF [28], and FGCN [29]. This may be due to other trade-offs made in achieving real-time performance, resulting in some data loss in other aspects.

The accuracy and loss of the first 60 epochs of model training are shown in Figure 4. From the figure, it can be observed that the loss is still decreasing, indicating that the EarlyStopping callback function has not been invoked at this point, and the training process will continue. To complete the analysis of the results, many studies use the F1-score in addition to accuracy [30]. After the completion of training, the F1-score on the SEED dataset’s test set is 0.854, demonstrating that the model has achieved a good balance between accuracy and recall. This suggests that the model is capable of correctly identifying samples from the three categories while also minimizing the occurrence of excessive misclassifications.

The confusion matrix is shown in Figure 5. From this, we can observe that the recognition rate for neutral emotion is the highest, reaching 90.8%, followed by negative emotion at 82.9%, and the recognition rate for positive emotion is the lowest, at only 82.4%. The recognition rate of neutral emotions is higher than that of positive and negative emotions, which may be related to the high proportion of neutral emotion samples in the data, or the model has better recognition ability for neutral emotion features.

The experimental results when using the SEED-IV dataset are shown in Table 2. The performance of the model is compared with SVM [31], BiDANN [32], and BiHDM [33]. SVM is a classical machine learning method, while the other methods are more advanced. In SVM, the EEG features are directly input into the support vector machine for emotion state prediction. In the graph-based methods, the brain features are precomputed before inputting into the network. From Table 2, it can be observed that our model achieves relatively high accuracy compared to both SVM and the graph-based methods, reaching 74.26%, which is slightly lower than the 74.35% accuracy of BiHDM.

The training process of the model is shown in Figure 6, displaying the accuracy and loss. From the figure, it can be observed that the loss has started to fluctuate, and the EarlyStopping function was invoked shortly thereafter, leading to the termination of training. After completing the training, the model achieved an F1-score of 0.707 on the SEED dataset during testing, indicating a favorable comprehensive performance of the model with room for improvement. In the future, the model will be further refined through methods such as adjusting model parameters, utilizing more complex network structures, and increasing the training dataset.

The confusion matrix is shown in Figure 7. From this, it can be observed that the model performs well in recognizing fear emotion, achieving an accuracy of 78.1%. And, the following are sadness emotion and neutral emotion, while the recognition rate for happy emotion is the lowest, with an accuracy of only 71.3%. The higher recognition rate for fear emotion could be attributed to both the characteristics of the data themselves and the possibility that fear emotion exhibits more prominent features in the EEG data. This is an area worth investigating further. Different emotions may show distinct trends in EEG data, and if the features of fear emotion are more recognizable, it can be beneficial for predicting emotions during emotional transitions, thereby greatly advancing the development of real-time emotion recognition based on EEG.

Two sets of experiments were conducted on the SEED and SEED-IV datasets, where the SEED group achieved higher recognition accuracy due to having only three emotions. However, the accuracy of both sets of experiments did not reach the expected goals, indicating that applying the results in real-life scenarios still presents certain challenges. To improve the accuracy of the experiments, the first step is to find suitable datasets. The datasets used in the experiment consist of 60 s of samples, with each sample labeled with a single emotion. When partitioning the data into windows, the emotions assigned to the windowed data are directly based on the original emotion labels whether using 1-s or 3-s windows. However, the true emotions within the windowed data may not necessarily match the original labels, which significantly affects the accuracy of the results. Therefore, collecting suitable datasets is of necessity for future research. In addition, due to the limitations of current brainwave detection devices, emotion recognition can only be performed after data collection. In the future, there is a need to overcome these equipment limitations and achieve real-time emotion recognition while detecting, enabling the application of real-time emotion recognition in everyday life.

## 5. Conclusions

This paper elaborates on the real-time problems of EEG-based emotion recognition. To achieve real-time emotion recognition, Δt needs to be minimized, and each emotion in every segment of data should be accurately identified, rather than just the dominant emotion of that segment. This requires finding more appropriate lengths of the time windows for data segmentation and placing greater emphasis on the latest data during the recognition process, reducing the impact of the past data on the recognition results. Furthermore, we apply a model that combines LSTM and attention mechanisms for validation on the SEED datasets and SEED-IV datasets. The model focuses more on the latest data, making it more significant during the recognition process. Experimental results indicate the feasibility of the real-time EEG-based emotion recognition. However, the results did not meet the expected goals since there are potential losses in realizing real-time performance. The method used in this article requires signal preprocessing, which increases the workload and makes it impossible to apply in practice. Lai [34] proposes a new architecture for convolutional neural networks (CNNs) using EEG signals that does not require complex signal preprocessing, feature extraction, or feature selection stages. Reducing these stages can effectively improve efficiency and is one of the key points to consider in the future. Future work will involve exploring a more suitable length of time window and, based on that, predicting and categorizing emotional changes. The early exploration after this paper may focus on emotions with significant EEG data variation, such as fear and sadness. The experimental process induces subjects’ emotions to continually fluctuate between fear and sadness, leveraging significant trends in these changes to predict emotions during the transition phase. After achieving accurate predictions, emotions with smaller trend changes can be considered.

## Figures and Tables

**Figure 1 sensors-23-07853-f001:**
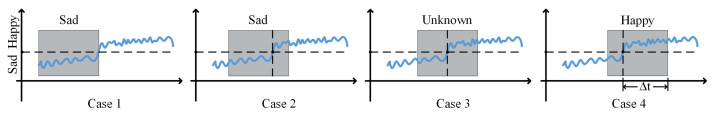
Recognition results under different proportion of new emotions.

**Figure 2 sensors-23-07853-f002:**
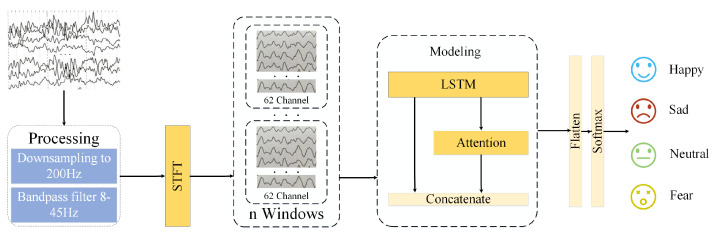
Framework for real–time EEG–based emotion recognition.

**Figure 3 sensors-23-07853-f003:**
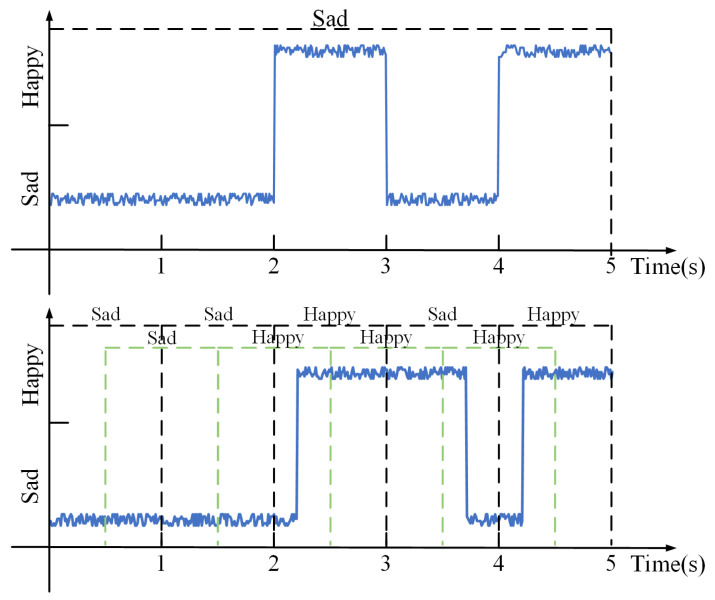
Recognition results with different time window lengths.

**Figure 4 sensors-23-07853-f004:**
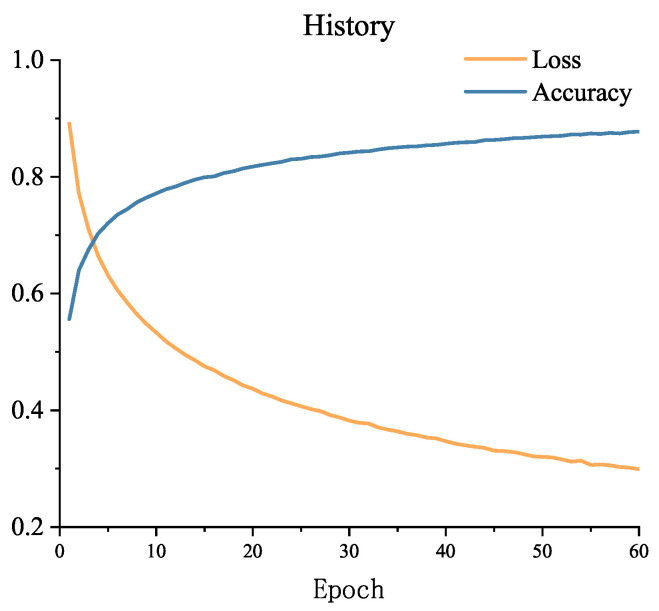
Model training loss changes with epoch on the SEED dataset.

**Figure 5 sensors-23-07853-f005:**
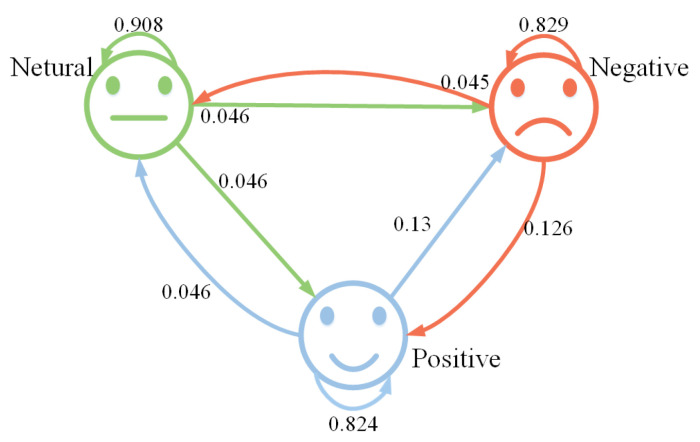
The confusion matrix of experimental results based on SEED dataset.

**Figure 6 sensors-23-07853-f006:**
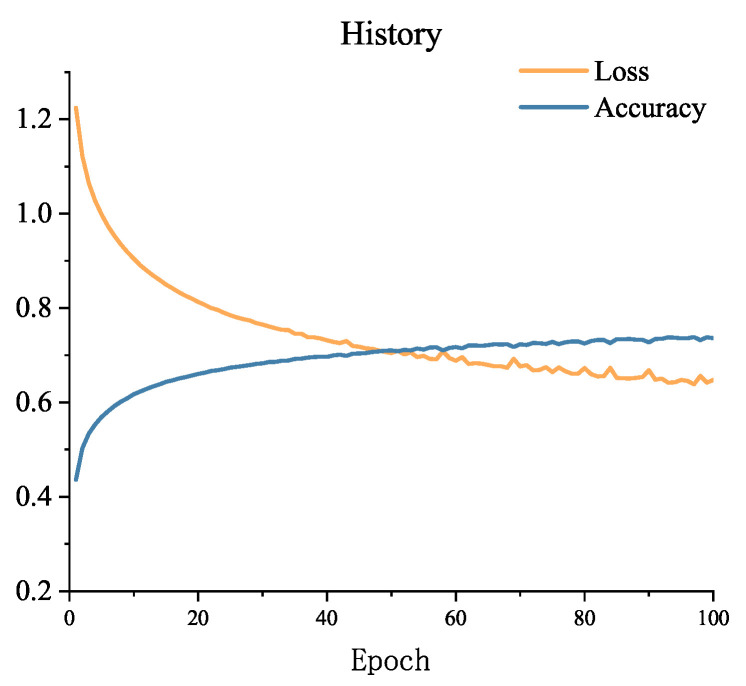
Model training loss changes with epoch on the SEED-IV dataset.

**Figure 7 sensors-23-07853-f007:**
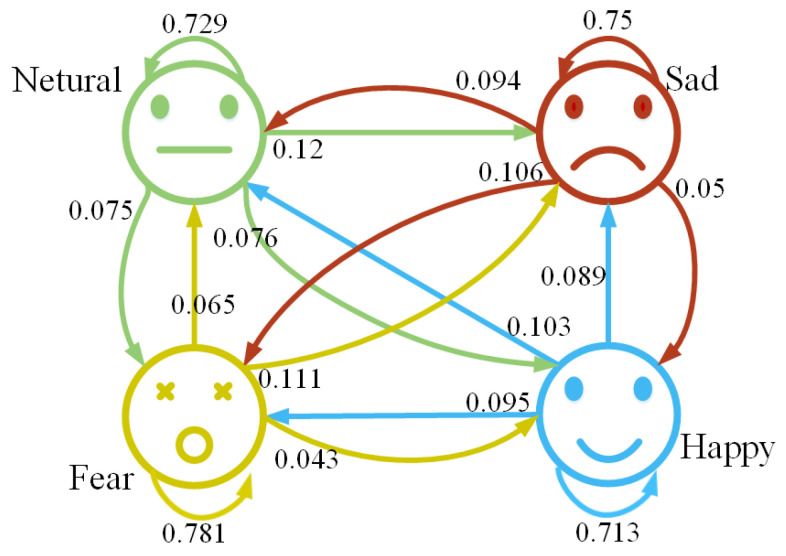
The confusion matrix of experimental results based on SEED-IV dataset.

**Table 1 sensors-23-07853-t001:** The model used was compared with other methods on the SEED dataset, and the best result is highlighted in bold.

Studies	Accuracy
KNN	72.60%
CNN	78.34%
DGCNN	90.40%
GELM	91.07%
BODF	93.80%
**FGCN**	**94.10%**
THIS WORK	85.40%

**Table 2 sensors-23-07853-t002:** The model used was compared with other methods on the SEED-IV dataset, and the best result is highlighted in bold.

Studies	Accuracy
SVM	56.61%
BiDANN	70.29%
**BiHDM**	**74.35%**
THIS WORK	74.26%

## Data Availability

Not applicable.

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
