# Peer review of "Real-Time EEG-Based Emotion Recognition"

_sensors, 2023, doi:10.3390/s23187853_

Round 1
Reviewer 1 Report
The goal of the presented research is to address the real-time performance problem in EEG-based emotion recognition. The main achievements are proposing a model combining LSTM and attention mechanism, using shorter time windows and focusing on latest data to capture emotional changes, and conducting experiments on two EEG emotion datasets to demonstrate the feasibility of real-time EEG emotion recognition achieving decent accuracy for real-time performance.
There are no major methodical or scientific issues. The methods and experiments are sound, but a couple potential issues:
The explanation of why shorter windows are better for real-time could be expanded.
More details could be provided on the model architecture and training process.
Overall the methodology seems scientifically sound. The use of shorter time windows and attention mechanism is logical for improving real-time performance. Feature extraction using STFT is a standard technique. The model combining LSTM and attention mechanism is reasonable for time series EEG data.
The quality of the language and writing could be improved in some areas. More details could be provided for some parts like the model architecture.
The research aims to address an important problem and provides a reasonable solution and results. More discussion of limitations and details would improve the quality.
E.g. the limitations of the work could be discussed in the conclusion. The accuracy is decent but still not ideal, so discussing limitations and future work is important.
The introduction could be more concise by removing some repetitive details when citing previous works.
Some figures are referenced out of order in the text.
The conclusion is a bit abrupt, can be expanded with some discussion of limitations and future work.
The research goal and methods are valid. Results are properly presented and discussed. Writing quality is good. Some minor improvements possible as mentioned in this review.
Section 4.3.2
line 319: "Table tab2" bad reference
line 327: "Figure fig7." bad reference
The order of publication details varies between references. For instance, the location of the date or year varies.
The use of punctuation is inconsistent. Some references use semicolons to separate authors, while others use commas.
Usage of "Vol." vs. "vol. " varies across references.
Some references miss the "pp." or "pp" when indicating page numbers. Using "pp." or "pp" varies across references.
There's inconsistency in the representation of volume and issue numbers. For example, references 4, 11, and 14 use "vol. XX, no. XX," while references 5, 6, and 12 just list the volume and issue numbers without any preceding text.
References 23 and 26 seem to be duplicates as they have the same authors, title, journal, volume, issue, and date.
In reference 5, strange abbreviation: "Ambient Intell Human Comput".
In reference 32, there's a minor typo with the location name "Shanghai, Chin" — it should be "Shanghai, China".
The representation of months is inconsistent. Some references use the full month name (e.g., "September" in reference 4), while others abbreviate it (e.g., "Sept" in reference 23 and 26), some while some references show full date details.
In a standardized reference list, typically one format is maintained throughout for a given type of source (journal, conference, book, etc.). Here, there's a mix of formats even for the same type of source.
The "[CrossRef]" or [PubMed] annotation is uniformly used throughout the list. However, the presence of these annotations in a standard reference list is unusual.
There are some minor grammatical issues, e.g.:
Introduction line 101: "reasearch" should be "research"
Section 4.1, line 252: "enhances" should be "enhance"
Section 5., line 350: "t needs" should be "it needs"
Author Response
Dear Reviewer,
I have addressed your comments point by point and incorporated them into the PDF document. Please see the attachment.
Best regards,
Xiangkun Yu

Reviewer 2 Report
This work is interesting. Emotion recognition is a relevant topic that is intensively investigated. There are different approaches for this recognition. In this paper, it is implemented based on EEG signal analysis. This study corresponds to the main issues of the journal.
In general, the work is presented clearly. The principal results indicated in the conclusion are discussed in detail, supported by the experimental study. There are enough figures and tables in the paper. They support the understanding of the paper sufficiently. Need to say about the comparison with other studies, which the authors discuss in section 4.3.2.
This comparison shows that the proposed result does not best accuracy in comparison with the other methods (Table 1). But for KNN and CNN classifiers, there is a significant effect of using shorter time windows. Therefore, EEG signals can be used in real-time emotion recognition. I'd like to recommend considering in the conclusion the use of a fuzzy classifier for the EGG signal, which allows an increase the efficiency of classification:
1. Rabcan, J., Levashenko, V., Zaitseva, E., Kvassay, M., Review of methods for EEG signal classification and development of new fuzzy classification-based approach, IEEE Access, 2020, 8, pp. 189720–189734
2. Lai, C.Q., Ibrahim, H., Suandi, S.A., Abdullah, M.Z., Convolutional Neural Network for Closed-Set Identification from Resting State Electroencephalography, Mathematics, 2022, 10(19), 3442
I would like to extend the evaluation of the result and in addition to Accuracy use other metrics such as Specificity, Sensitivity, Balanced accuracy, Precision, F1-score, Matthews correlation coefficient. Of particular note is the F1-score, which allows you to evaluate the effectiveness of classifiers induced based on unbalanced data. These metrics, in particular, are considered for the evaluation in [1].
Author Response

(The authors gave the same response as above.)

Round 2
Reviewer 1 Report
The authors properly addressed the reviewers' comments.
The quality of the paper has been improved.
There are still some minor issues, after correcting them I suggest the paper for publication.
Some of the minor issues can be read as follows.
Figure 4. The legend could be placed to right-middle to avoid overlap with the plot.
For the figures using vector graphic format (e.g. eps) can improve the quality.
Tables: "the best result is highlighted in bold" - I cannot see bold line in the table.
multiple occurrences: ’∆t’ - bad formatting
Indication of [CrossRef] and [PubMed] in the References section is not standard in the publication style of the journal.
Subsection 4.3 title "Result" should be "Results"
Some of the minor issues can be read as follows.
line 262: 2,62,128 - missing space after comma
line 372: "phase.After" - missing space after dot
Author Response
All modifications are shown in bold.
Point 1: Figure 4. The legend could be placed to right-middle to avoid overlap with the plot. For the figures using vector graphic format (e.g. eps) can improve the quality.
Response 1: Thanks for your review, the legend has been placed in the middle right and the image quality has been improved.
Point 2: Tables: "the best result is highlighted in bold" - I cannot see bold line in the table.
Response 2: Thanks for your suggestions, the best results are highlighted in bold.
Point 3: multiple occurrences: ’∆t’ - bad formatting
Response 3: Thanks for your comment, it has been edited.
Point 4: Indication of [CrossRef] and [PubMed] in the References section is not standard in the publication style of the journal.
Response 4: Thanks for your suggestion, all references have been re-hyperlinked.
Point 4: Subsection 4.3 title "Result" should be "Results".
Response 4: Thanks for your suggestion, it has been modified.
Point 5: line 262: 2,62,128 - missing space after comma
line 372: "phase.After" - missing space after dot
Response 5: Thanks for the suggestion, spaces have been added.
Reviewer 2 Report
I thank the authors for the consideration of my comments. The metrics F1 score shows the efficiency of the proposed method.
The advantages and disadvantages of the proposed method should be indicated in the conclusion. I’d like to recommend including in the conclusion the possible ways for future studies, taking into account, the modification of procedures of feature selections and other types of classifiers. The authors can analyze the advantages and disadvantages of these possible ways based on published studies.
Author Response
All modifications are shown in bold.
Thanks for the suggestion, showing in published studies that the F1 score allows to evaluate the effectiveness of classifiers induced based on imbalanced data, analyzing the advantages of possible methods based on published studies and pointing out the shortcomings of the proposed methods in the conclusion.